# Factors Influencing Pregnant Women’s Injuries and Fetal Loss Due to Motor Vehicle Collisions: A National Crash Data-Based Study

**DOI:** 10.3390/healthcare9030273

**Published:** 2021-03-03

**Authors:** Shinobu Hattori, Masahito Hitosugi, Shingo Moriguchi, Mineko Baba, Marin Takaso, Mami Nakamura, Seiji Tsujimura, Yasuhito Miyata

**Affiliations:** 1Department of Legal Medicine, Shiga University of Medical Science, Tsukinowa, Seta, Otsu, Shiga 520-2192, Japan; shattori@fujita-hu.ac.jp (S.H.); opi717@belle.shiga-med.ac.jp (S.M.); marint@belle.shiga-med.ac.jp (M.T.); mamin@belle.shiga-med.ac.jp (M.N.); 2Center for Integrated Medical Research, Keio University School of Medicine, 35 Shinanomachi, Shinjuku-ku, Tokyo 160-8582, Japan; mineko@keio.jp; 3Joyson Safety Systems Japan K.K. Echigawa Plant, 658 Echigawa, Aisho-cho, Echi-gun, Shiga 529-1388, Japan; Seiji.Tsujimura@jp.joysonsafety.com (S.T.); Yasuhito.Miyata@jp.joysonsafety.com (Y.M.)

**Keywords:** pregnant women, motor vehicle collision, passenger, injury severity, fetus fatality

## Abstract

To examine the factors that influence substantial injuries for pregnant women and negative fetal outcomes in motor vehicle collisions (MVCs), a retrospective analysis using the National Automotive Sampling System/Crashworthiness Data System was performed in Shiga University of Medical Science. We analyzed data from 736 pregnant women who, between 2001 and 2015, had injuries that were an abbreviated injury scale (AIS) score of one or more. The mean age was 25.9 ± 6.4 years and the mean gestational age was 26.2 ± 8.2 weeks. Additionally, 568 pregnant women had mild injuries and 168 had moderate to severe injuries. Logistic regression analysis revealed that seatbelt use (odds ratio (OR), 0.30), airbag deployment (OR, 2.00), and changes in velocity (21–40 km/h: OR, 3.03; 41–60 km/h: OR, 13.47; ≥61 km/h: OR, 44.56) were identified as independent predictors of having a moderate to severe injury. The positive and negative outcome groups included 231 and 12 pregnant women, respectively. Injury severity in pregnant women was identified as an independent predictor of a negative outcome (OR, 2.79). Avoiding moderate to severe maternal injuries is a high priority for saving the fetus, and education on appropriate seatbelt use and limiting vehicle speed for pregnant women is required.

## 1. Introduction

Worldwide, the number of motor vehicle collisions (MVCs) has been increasing continuously, with 1.35 million associated fatalities in 2016 [1]. MVCs are predicted to become an even bigger problem by 2030, when they are projected to be the fifth most common cause of fatalities in the world [1]. Therefore, greater efforts by all countries are recommended to decrease MVC fatalities. A recent systematic review of trauma in pregnant women showed that a MVC is the most common and the most life-threatening mechanism of injury [2].

In the United States, state-level linkage studies have estimated that the pregnancy crash risk among pregnant front-seat occupants or drivers ranges from 1.0% to 2.8% [3,4,5]. In Japan, although the data in this area are unknown because of the lack of nationwide statistics, 2.9% of pregnant women were involved in MVCs [6]. Reliable statistics on fetal loss due to MVCs are not available because maternal involvement in crashes has not been consistently recorded on the fetal death certificate, and it has been estimated that the number of fetal losses may be greater than the number of infant deaths due to MVCs [7]. Therefore, great efforts are required to save the life of both mothers and fetuses that are involved in MVCs. Protecting the mother’s life is the first step toward preventing fetal death due to MVCs.

Generally, as a tool of evaluating anatomical injury severity, abbreviated injury severity (AIS) scores have been widely used [8]. The AIS score is used to categorize the injury type and severity anatomically in each body region on a scale from one (minor) to six (clinically untreatable). We have previously found that a negative fetal outcome occurs when the mother has minor injuries, with an abbreviated injury scale (AIS) score of one, and that predicting the fetus’s outcome on the basis of the mother’s injury severity was difficult [9]. Furthermore, a negative fetal outcome can occur in mothers without any anatomical injuries [10]. Therefore, to establish strategies for saving fetuses, we have to understand the risks for the mother’s substantial injuries and a negative fetal outcome for those that are involved in MVCs.

For patients involved in MVCs, there were several factors about the collision characteristics that influenced the mechanisms of injuries and outcomes. Collision details have to be considered in the study of pregnant vehicle passengers who are involved in a collision. Previously, a nationwide hospital-based database was analyzed, and the factors for pregnancy loss or for requiring surgery were determined [11,12,13]. Additionally, national or regional population-based databases have been used to study pregnant women who were involved in MVCs [3,14,15,16,17]. However, these databases lack detailed information on vehicle collisions. Therefore, the NASS/CDS database, which includes information about the crash circumstances and scene, have been used to determine the relationship between the crash severity and pregnant women’s injuries or outcomes. Previously, several studies on pregnant women who were involved in MVCs were performed using this database or similar in-depth investigations [10,18,19]. Klinich et al. suggested that a greater crash severity, more severe maternal injuries, and a lack of proper seatbelt use were associated with an adverse fetal outcome with in-depth investigations of crashes. However, this study was based on small numbers of collisions involving 57 pregnant occupants, and factors about the principal direction of the force, action of pretensioner system, or rollover were not included in the analysis [18]. Manoogian compared the crash and injury characteristics between pregnant and non-pregnant vehicle occupants and found that the risk of injuries with an AIS score of two or more for pregnant occupants was similar to the risk of those for non-pregnant occupants [10]. Collins et al. compared the restraint use rate between pregnant and non-pregnant women who were involved in MVCs, and they concluded that pregnant women wear seatbelts at significantly lower frequencies than non-pregnant women [19]. However, no study has determined the factors that influence more severe injuries for pregnant women and a negative fetal outcome in MVCs where the crash details were included.

The main objective of this study was to examine the factors that influence substantial injuries for pregnant women and a negative fetal outcome in MVCs using a national crash database to establish effective preventive measures.

## 2. Materials and Methods

### 2.1. Study Design

This observational study was a retrospective analysis using the National Automotive Sampling System/Crashworthiness Data System (NASS/CDS). The NASS/CDS provides nationally representative detailed data of approximately 5000 MVCs, which have been investigated each year, involving cars, light trucks, vans, and sport utility vehicles. To be recorded in the database, at least one of the vehicles involved in the collision must have been damaged enough to require that it was towed from the scene. The database comprised data that were collected from interviews for involved persons, police records, medical records, vehicle inspections, scene inspections, and photographs. This database has been open to the public and anyone can access it. The raw data were downloaded by some of the authors (ST and YM) via the FTP site of the NASS/CDS in July 2017 [20]. The following analyses were performed in Shiga University of Medical Science. Because of the anonymous and retrospective nature of this study, the need for informed consent was waived.

### 2.2. Patient Selection

A total of 65,390 collisions involving 141,057 persons were registered in the NASS/CDS from 2001 to 2015. Among these collisions, we included cases that involved at least one pregnant occupant. Subsequently, 1074 collisions involving 1088 pregnant women were collected. The 352 pregnant women who had no anatomical injuries, indicating an AIS score of 0, were excluded from analyses. Finally, 736 pregnant women who experienced injuries with an AIS score of 1 or more were selected for analyses (Figure 1). This screening procedure was performed by some of the authors (MH, SM, ST, and YM).

### 2.3. Collected Data

The database included the variety of information regarding the involved pregnant women (i.e., age, stature, gestational age, type, and severity of injuries) and the collisions (i.e., seating position, direction, and velocity of the collision, use or acting of safety systems). From the information, we chose the data considered influential to the mechanism of injuries of the pregnant woman. Subsequently, the following information was obtained from the database for each victim:

(1) General subject characteristics: age, height, and weight.

(2) Trimester period (gestational age: <13 weeks, first trimester; 13 to 27 weeks, second trimester; and >27 weeks, third trimester).

(3) Location in the vehicle (left front, right front, or rear).

(4) Seatbelt use and action of the pretensioner system.

(5) Airbag deployment.

(6) Number of collisions (how many times the vehicle collided with other vehicles or objects).

(7) Principal direction of the force. Clock directions were used in degrees for the principal force direction that resulted in the highest number of crashes. For example, the front of the vehicle was 0° or 360° and the rear of the vehicle was 180°. Thus, the following definitions were applied: frontal, 330° to 30°; right lateral, 31° to 149°; rear, 150° to 210°; and left lateral, 221° to 329°).

(8) Total changes in vehicle velocity (delta-V total (DVTOTAL)). DVTOTAL is combined lateral and longitudinal delta-V, which was determined using the NASS/CDS. The values were rated in multiples of 10 km/h.

(9) Rollover.

(10) Occupant injury severity described using the AIS score. If the pregnant women had multiple injuries in the same region, the maximal score was shown. The maximum AIS (MAIS) score was defined as the highest AIS value for all body regions in each pregnant woman.

(11) Outcomes for the pregnant women and fetus. The outcome was defined as alive or dead. Fetal outcomes were examined for cases from 2008. Fetal death was defined as death within 1 month after the MVC.

### 2.4. Statistical Analysis

Data were summarized in the form of values with proportions or frequencies for categorical variables. The mean ± standard deviation for the values that followed a normal distribution and the median and interquartile range (IQR) for values that did not follow a normal distribution were used to summarize continuous variables. Chi-square tests were used to compare the prevalence between two groups. To determine if there was a significant difference between the means of two groups, Student’s *t*-test was used. To find the differences in the values without a normal distribution, the Mann–Whitney *U*-test was conducted. A *p* value of 0.05 or less was considered to be statistically significant. To identify variables that were independently associated with substantial injuries to the pregnant women or a negative fetal outcome, we performed a logistic regression analysis. The analyses were performed using SPSS ver. 23 (IBM, Chicago, IL, USA).

## 3. Results

### 3.1. Participant Demographic Information

For 736 pregnant women involved in 728 collisions, the mean age was 25.9 ± 6.4 years. The average height was 164.0 ± 8.3 cm and the average weight was 74.6 ± 19.1 kg. The mean gestational age was 26.2 ± 8.2 weeks. Two hundred twenty-four pregnant women (30.4%) were in the first trimester, 258 (35.1%) were in the second trimester, 229 (31.1%) were in the third trimester, and 25 were unknown. Additionally, 408 (55.3%) pregnant women were primipara (mean age, 30.1 ± 5.1 years) and 330 (44.7%) were multipara (mean age, 33.0 ± 4.4 years).

### 3.2. Collision Characteristics

Four hundred seventy-two pregnant women (64.1%) were seated in the left-front seat (suggesting a driver), 206 (28.0%) were in the right-front seat (suggesting a front passenger), and 48 (6.5%) were rear-seat passengers. Seventy percent of pregnant women were belted and 25.1% were unbelted. The mean number of collisions was 1.6 ± 1.0. Frontal collision was the most common (45.4%) followed by left lateral collision (12.2%), right lateral collision (11.1%), and rear collision (6.8%). For the DVTOTAL distribution, 11–20 km/h was the most common (22.7%) followed by 21–30 km/h (15.5%), 31–40 km/h (8.7%), and 41–50 km/h (5.3%). For 41.8% of pregnant women, the airbag was deployed in front of their seat during the collision.

### 3.3. Fetal Injury Severity and Outcome

The distribution of the MAIS scores is shown in Figure 2. Most of the pregnant women (77.2%) had mild injuries with a MAIS score of one. For cases between 2008 and 2015, we investigated the outcome for pregnant women and fetuses. We found that 231 pregnant women and their fetuses were alive; three pregnant women and their fetuses died; nine pregnant women were alive, but their fetuses died; and two pregnant women died, but their fetuses were alive.

### 3.4. Comparison of Mild or Moderate to Severe Injuries

We divided pregnant women into groups as follows: mild injuries with a MAIS score of one and moderate to severe injuries with a MAIS score of two or more. The pregnant women’s backgrounds and collision characteristics were compared between these two groups (Table 1). There were no significant differences in the pregnant women’s backgrounds. Pregnant women with moderate to severe injuries had a significantly higher velocity of collision, frequently used a seatbelt, and frequently had airbag deployment, which was significantly different from the mild injury group (*p* < 0.001).

Next, to identify the variables that were independently associated with moderate or severe injuries, logistic regression analysis was performed on the basis of the univariate analysis results. Seatbelt use, airbag deployment, and DVTOTAL showed significant differences in the univariate analyses, and these factors were included in the logistic regression analysis. Ultimately, seatbelt use (odds ratio (OR), 0.30), airbag deployment (OR, 2.00), DVTOTAL (21–40 km/h: OR, 3.03; 41–60 km/h: OR, 13.47; ≥61 km/h: OR, 44.56) were identified as independent predictors of having a moderate to severe injury (Table 2).

### 3.5. Comparison of Fetal Outcomes

We divided the outcomes into positive (both pregnant women and fetuses were alive) and negative (fetal death). The positive and negative outcome groups included 231 and 12 pregnant women, respectively. Pregnant women’s backgrounds and collision characteristics were compared between the two groups (Table 3). There were no significant differences in the pregnant women’s background between the two groups. Pregnant women with a negative fetal outcome had significantly more frequent left-side collisions (*p* = 0.019) and a higher MAIS score (*p* < 0.001) compared with the positive outcome group.

To identify variables that were independently associated with a negative outcome, logistic regression analyses were then performed on the basis of the univariate results. The principal direction of the force and the MAIS score, which showed significant differences in the univariate analyses, were included in the logistic regression analysis. Ultimately, the MAIS score in pregnant women was identified as an independent predictor of a negative outcome (OR, 2.79; Table 4).

## 4. Discussion

In this study, we found that the MAIS score for a pregnant woman was an independent predictor of a negative fetal outcome. Therefore, decreasing the maternal injury severity is the highest objective. A systematic review suggested that high severity trauma was an independent risk factor for immediate complications in pregnant women, and a high injury severity score (ISS), which is the sum of the squares of the highest AIS scores in each of the three most severely injured body regions, is also associated with late-term complications including preterm labor, placental abruption, and perinatal morbidity [2]. However, a retrospective analysis of pregnant women who were admitted to hospital after MVCs suggested that the ISS might not be a predictive risk factor for a negative pregnancy outcome including fetal loss because there was no significant correlation between the ISS and an immediate adverse maternal and fetal outcome [11]. Schiff and Holt also reported that the ISS was not accurate for predicting placental abruption and fetal death [16]. These theories resulted from the difficulty in predicting fetal outcome for maternal injuries with a low severity. As suggested previously, a negative fetal outcome occurred when the mother had minor injuries or no anatomical injuries [2,9]. More severe maternal injuries are considered to be an independent risk factor for fetal mortality [12,21]. Although different conclusions have been reported regarding the maternal injury severity and fetal outcome, our results confirmed that elevated maternal injury severity was well associated with a negative fetal outcome.

In this study, it was of great interest that wearing a seatbelt was a negative while a higher collision speed was a positive influence on having moderate to severe injuries, but they did not significantly influence the fetal outcome. These issues were a novelty of this study and, regarding the seatbelt use, the present result was different from the previous reports. For the collision speed, a study using in-depth crash data classified the crash severity using delta-V as follows: >48 km/h, severe; 24–48 km/h, moderate; and <24 km/h, minor. They concluded that the crash severity was a significant predictor of fetal outcome [18]. However, the value of the OR was low (1.1). Therefore, because collision speed was not a significant predictor of fetal outcome in this study, we considered that our results were similar to those of previous studies. Prospective analyses or a similar study that includes more cases with a negative fetal outcome may confirm the present result. For appropriate seatbelt use, a systematic review of trauma in pregnancy suggested that the major risk factor for adverse fetal outcome following MVC was improper seatbelt use [22]. Additionally, the study that used in-depth crash data concluded that improper restraint was a significant predictor of a negative fetal outcome [18]. The differences between the results of that study and the present study might be due to differences in the sample size. As previously mentioned, our study included a larger number of cases and took into account more indices that were related to the collision. Thus, the authors believe that more reliable results were obtained in the present study. An additional reason that seatbelt use was not selected as a significant predictor for a negative fetal outcome was the occurrence of a negative fetal outcome for properly belted pregnant women. It has been suggested that negative fetal outcomes may result from restrained pregnant woman passengers who are involved in minor vehicle collisions. One report found that among fetal losses that were associated with trauma, 60–70% of pregnant women had minor injuries [23]. One retrospective cohort study determined that even when pregnant women wore a seatbelt, preterm births occurred in 122 of 100,000 pregnancy days [17]. The same report observed 5.2 stillbirths, 7.0 placental abruptions, and 22.3 premature ruptured membranes [17]. A biomechanical study using a pregnant woman dummy revealed that when properly restrained, the chest was deflected 35.4 mm and may compress the enlarged uterus in a frontal collision at 26 km/h [24]. The study also suggested that, if the pregnant occupant was properly restrained, a negative fetal outcome might occur because of chest compression and subsequent forces that are applied to the uterus. Therefore, further study is required to confirm whether proper seatbelt use is a significant predictor of a negative fetal outcome.

Because the MAIS score in pregnant women was the only significant predictor of a negative outcome, avoiding moderate to severe maternal injuries is a high priority. Proper seatbelt use and decreasing the vehicle velocity at collision, as shown in our results, were effective measures to prevent these maternal injuries. Additionally, a biomechanical study using a pregnant woman dummy confirmed that even in a low-speed vehicle collision (13, 26, and 40 km/h), the unrestrained pregnant woman driver had severe or fatal injuries [25]. Recently, although a higher rate of seatbelt use in pregnant woman passengers has been observed in developed countries, a substantial number of these women use seatbelts incorrectly [26,27,28,29,30]. A report in Japan showed that, although most pregnant woman drivers (97.6%) always wore a seatbelt, 12.7% of them did so incorrectly [30], and a survey in Alabama, USA suggested that the shoulder or lap belt was incorrectly used by 27.5% of pregnant women [29]. Therefore, health-care professionals should provide proper counselling about correct seatbelt use for pregnant women.

Although road safety is influenced by many factors, it has been suggested that the effects of some variables on safety are mediated by vehicle speed [31]. A prospective study on high-energy MVCs that resulted in immediate hospitalization of younger drivers suggested that among culpable drivers, speeding behavior was the main predisposing factor to the collisions [32]. Our results indicated that speeding is a risk factor for the moderate and severe pregnant women’s injury at collision. The National Highway Traffic Safety Administration and the Insurance Institute for Highway Safety proposed the Automatic Emergency Braking Initiative in 2015, which is intended to make automatic emergency braking with forward collision warning systems standard on nearly all new cars by September 2022 [33]. Further development of safety systems may reduce the collision speed in MVCs and contribute to the reduction of moderate and severe injuries to vehicle passengers. However, until the precrash safety technology is implemented, safety education to prevent excessive speeding, especially for pregnant women, is required.

There are some limitations in this study. First, the NASS/CDS includes a sample of crashes for which at least one of the crash-involved vehicles was towed away from the collision scene. Therefore, less severe crashes are not included in the database. However, our objectives were to determine the factors that influence substantial injuries for pregnant women and a negative fetal outcome in MVCs. Because moderately and severely injured pregnant women and those with a negative fetal outcome were included in this study, the authors suggest that this limitation did not influence the results. Second, although this database consisted of huge samples, the number of cases with a negative fetal outcome was as low as 12 between 2001 and 2015. Furthermore, as described above, there may be some pregnant women with a negative fetal outcome after having minor traffic injuries. Further in-depth studies to collect data on pregnant women with a negative fetal outcome due to MVCs are required in the future. Third, this study was based on a single database of vehicle crashes in the United States. Because this database does not reflect the traffic environment and characteristics in other countries, e.g., different body size, vehicle size, and speed limit, future research should compare real-world crash data involving pregnant women using international sources.

## 5. Conclusions

To save the fetus, avoiding moderate to severe maternal injuries is a high priority. We health-care professionals should provide proper counselling about correct seatbelt use and safety education to prevent excessive speed for pregnant women. These measures may improve fetal outcomes if pregnant women are involved in MVCs. Also, measures may relieve the anxiety of fetal loss, and subsequently, promote the social participation of pregnant women.

## Figures and Tables

**Figure 1 healthcare-09-00273-f001:**
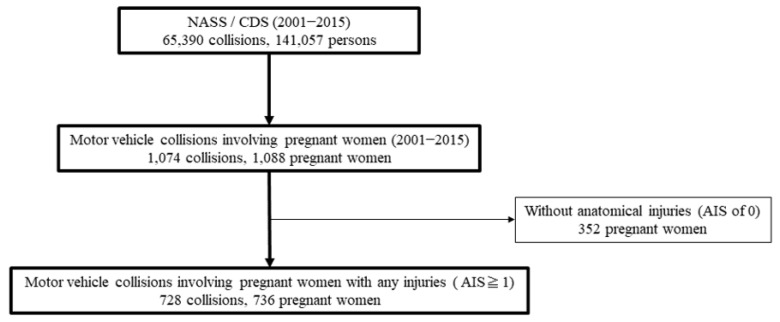
Flowchart of patient enrolment.

**Figure 2 healthcare-09-00273-f002:**
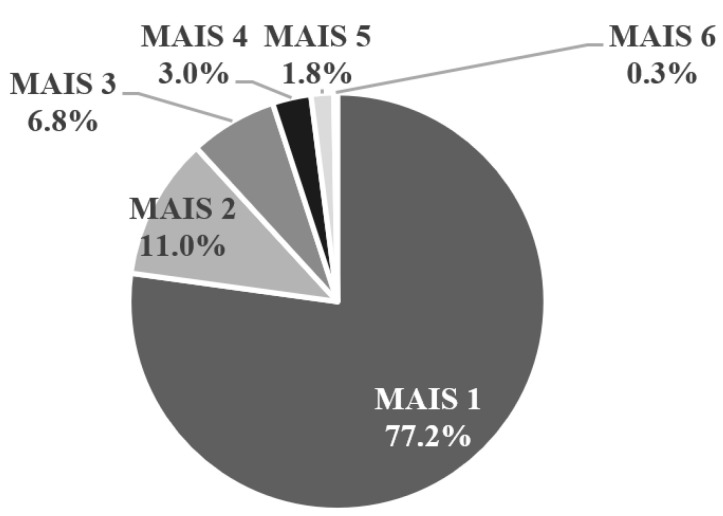
The maximum abbreviated injury severity (MAIS) score distribution for all pregnant women.

**Table 1 healthcare-09-00273-t001:** Comparison of the background and collision characteristics of pregnant women with a MAIS of one (n = 568) or two or more (n = 168).

Item	MAIS 1 (n = 568)	MAIS 2+ (n = 168)	*p* Value
Age (yr)	25.9 ± 6.1	26.0 ± 7.5	0.830
Height (cm)	163.7 ± 8.5	164.9 ± 7.7	0.143
Weight (kg)	74.7 ± 18.7	74.1 ± 20.7	0.754
Seatbelt use			<0.001
Yes	432 (76.1%)	83 (49.4%)	
No	116 (20.4%)	69 (41.1%)	
Unknown	20 (3.5%)	16 (9.5%)	
Pretensioner			0.837
Acting	71 (12.5%)	20 (11.9%)	
Not acting *	497 (87.5%)	148 (88.1%)	
Airbag deployment			0.009
Yes	223 (39.3%)	85 (50.6%)	
No	345 (60.7%)	83 (49.4%)	
Number of collision	1.6 ± 1.0	1.8 ± 1.1	0.857
Seating position			0.316
Front left	373 (65.7%)	103 (61.3%)	
Front right	151 (26.6%)	55 (32.7%)	
Rear	44 (7.7%)	10 (6.0%)	
Direction of forces			0.084
Front	260 (45.8%)	74 (44.0%)	
Rear	45 (7.9%)	5 (3.0%)	
Left	64 (11.3%)	26 (15.5%)	
Right	64 (11.3%)	18 (10.7%)	
Unknown	135 (23.7%)	45 (26.8%)	
Delta-v total (DVTOTAL) (km/h)			<0.001
1–20	180 (31.7%)	14 (8.3%)	
21–40	137 (24.1%)	41 (24.4%)	
41–60	21 (3.7%)	28 (16.7%)	
61–	2 (0.4%)	8 (4.8%)	
Unknown	228 (40.1%)	77 (45.8%)	
Rollover			0.276
Yes	69 (12.1%)	25 (14.9%)	
No	494 (87.0%)	136 (81.0%)	
Unknown	5 (0.9%)	7 (4.1%)	

* Including unknown whether acting and unknown whether equipped.

**Table 2 healthcare-09-00273-t002:** Results of logistic regression analysis to predict moderate to severe injuries in pregnant women.

Variable	Odds Ratio	95% Confidence Interval	*p* Value
Seatbelt use	0.304	0.172–0.536	<0.001
Airbag deployment	2.002	1.137–3.523	0.016
DVTOTAL			
1–20 (Ref)	N/A	N/A	N/A
21–40	3.030	1.555–5.904	0.001
41–60	13.469	5.986–30.304	<0.001
61–	44.564	8.118–244.618	<0.001

N/A: Not applicable.

**Table 3 healthcare-09-00273-t003:** Background and collision characteristics of pregnant women with a positive or negative fetal outcome.

Item	Both Alive (n = 231)	Fetal Death (n = 12)	*p* Value
Age (yr)	25.6 ± 5.7	28.1 ± 6.5	0.145
Height (cm)	164.2 ± 7.1	164.1 ± 6.5	0.878
Weight (kg)	75.7 ± 20.4	79.6 ± 20.8	0.427
Period of trimester			0.481
First	63 (27.3%)	4 (33.3%)	
Second	95 (41.1%)	3 (25.0%)	
Third	67 (29.0%)	5 (41.7%)	
Unknown	6 (2.6%)	0 (0%)	
Seatbelt use			0.670
Yes	172 (74.4%)	8 (66.7%)	
No	48 (20.8%)	3 (25.0%)	
Unknown	11 (4.8%)	1 (8.3%)	
Pretensioner			0.697
Acting	65 (28.1%)	4 (33.3%)	
Not acting *	166 (71.9%)	8 (66.7%)	
Airbag deployment			0.279
Yes	117 (50.6%)	8 (66.7%)	
No	114 (49.4%)	4 (33.3%)	
Number of collision	1.7 ± 1.0	1.7 ± 0.8	0.970
Seating position			0.436
Front left	155 (67.1%)	10 (83.3%)	
Front right	60 (26.0%)	2 (16.7%)	
Rear	16 (6.9%)	0 (0%)	
Direction of forces			0.019
Front	114 (49.4%)	4 (33.3%)	
Rear	28 (12.1%)	0 (0%)	
Left	15 (6.5%)	0 (0%)	
Right	22 (9.5%)	4 (33.3%)	
Unknown	52 (22.5%)	4 (33.3%)	
DVTOTAL (km/h)			0.374
1–20	57 (24.7%)	3 (25.0%)	
21–40	48 (20.8%)	3 (25.0%)	
41–60	15 (6.5%)	2 (16.7%)	
61–	3 (1.3%)	1 (8.3%)	
Unknown	108 (46.7%)	3 (25.0%)	
Rollover			0.771
Yes	25 (10.8%)	1 (8.3%)	
No	202 (87.5%)	11 (91.7%)	
Unknown	4 (1.7%)	0 (0%)	
MAIS	1 (1, 1)	3 (1, 3.8)	<0.001

* Including unknown whether acting and unknown whether equipped.

**Table 4 healthcare-09-00273-t004:** Results of the logistic regression analysis to predict a negative fetal outcome.

Variable	Odds Ratio	95% Confidence Interval	*p* Value
Direction of forces			
Front (Ref)	N/A	N/A	N/A
Rear	0.000	N/A	0.999
Left	3.838	0.726–20.292	0.113
Right	0.000	N/A	0.998
MAIS	2.787	1.589–4.890	<0.001

N/A: Not applicable.

## Data Availability

The data presented in this study are available upon request from the corresponding author.

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
