# Peer review of "Factors Influencing Pregnant Women’s Injuries and Fetal Loss Due to Motor Vehicle Collisions: A National Crash Data-Based Study"

_healthcare, 2021, doi:10.3390/healthcare9030273_

Round 1

Reviewer 1 Report

1) Your introduction is very short and only contains 9 citations. Please, expand it. For example, you could explain how the AIS is calculated. Don’t necessarily make your reader go and read other publications just to understand yours.

2) You have two groups, MAIS 1 and MAIS 2+, of different sample sizes. You should perform statistical analyses to compared the two. For example, ‘a t-test is a type of inferential statistic used to determine if there is a significant difference between the means of two groups.’. You even mention that, ‘To find the differences in the values between two groups, the Student’s t-test was used […]’, but, then, nowhere in the paper you mention it again, no ’t’ values can be found.

3) I appreciate that you discussion section is long, but perhaps some of it belongs in the introduction and/or conclusion?

4) And, then, the conclusion is too short, again. What are the issues with your study?

5) How are your results compared with other similar studies?

6) What could be done to use these results to better the society?

7) Shouldn’t there be IRB when using these data?

Author Response

Thank you for the thoughtful and constructive feedback you provided regarding our manuscript. In accordance with your suggestion, we have revised the manuscript. We are grateful for the time and energy you expended on our behalf

1) Your introduction is very short and only contains 9 citations. Please, expand it. For example, you could explain how the AIS is calculated. Don’t necessarily make your reader go and read other publications just to understand yours.

  Following to the reviewer’s suggestion, we have expanded the descriptions of Introduction. Part of the descriptions in Discussion have moved to the Introduction. Also, the explanation of scoring AIS have added to the Introduction (Line 50-53, 60-84).

2) You have two groups, MAIS 1 and MAIS 2+, of different sample sizes. You should perform statistical analyses to compared the two. For example, ‘a t-test is a type of inferential statistic used to determine if there is a significant difference between the means of two groups.’. You even mention that, ‘To find the differences in the values between two groups, the Student’s t-test was used […]’, but, then, nowhere in the paper you mention it again, no ’t’ values can be found.

  Following to your suggestion, we have revised the descriptions of Statistical analysis as follows (Line 150-153): “To determine if there is a significant difference between the means of two groups, the Student’s t-test was used. To find the differences in the values without a normal distribution, the Mann–Whitney U-test was conducted”.

3) I appreciate that you discussion section is long, but perhaps some of it belongs in the introduction and/or conclusion?

  In accordance with your suggestion, first paragraph of the Discussion have moved to the Introduction and revised the descriptions of Discussion.

4) And, then, the conclusion is too short, again. What are the issues with your study?

  I appreciate your suggestion and following suggestion by Reviewer 2: “Do not repeat your method and some key findings in the conclusions section. In the conclusions section, researchers may give some policy measures on this issue and suggest some recommendations in this section as well”. Therefore, we have revised the Conclusion as follows (Line 327-332): “To save the fetus, avoiding moderate to severe maternal injuries is a high priority. We health-care professionals should provide proper counselling about correct seatbelt use and safety education to prevent excessive speed for pregnant women. These measures might improve the fetal outcome if pregnant women are involved in MVCs. Subsequently, the measures also promote the social participation of pregnant women with relieving the anxiety of fetal loss”.

5) How are your results compared with other similar studies?

  In accordance with your suggestion, we have added the descriptions that the difference or similarity of the present results to the previous results. Following sentences have added in Discussion (Line 246-249, Line 252-254): “Although different conclusions have been reported regarding the maternal injury severity and fetal outcome, our result confirmed that elevated maternal injury severity was well associated with the negative fetal outcome”; “These issues were novelty of this study and regarding the seatbelt use, present result was different from the previous reports”.

6) What could be done to use these results to better the society?

  According to your suggestion, following sentences have included in Conclusion (Line 327-332): “We health-care professionals should provide proper counselling about correct seatbelt use and safety education to prevent excessive speed for pregnant women. These measures might improve the fetal outcome if pregnant women are involved in MVCs. Subsequently, the measures also promote the social participation of pregnant women with relieving the anxiety of fetal loss”.

7) Shouldn’t there be IRB when using these data?

  The NASS/CDS database has been open to the public and anyone can access it. Furthermore, this database was constructed with unlinkable anonymization. Therefore, there was no need to undergo ethical review. We have added the following sentences in Study design (Line 100-101): “This database has been open to the public and anyone can access it”; “Because of anonymous and retrospective nature of this study, the need for informed consent was waived”.

Reviewer 2 Report

It is a pleasure to review this important work on Factors influencing pregnant women’s injuries and fetal loss due to motor vehicle collisions: a national crash data-based study. 

Abstract

Neither title nor abstract reporting the site of research. It is important to mention the site in the abstract for a quick understanding of readers. The site can be added in Line#16-17

 Introduction

It would be great to mention/cite some previous studies that focus on factors influencing pregnant women’s injuries and fetal loss due to motor vehicle collisions. If there is no study on this topic, then clearly mention the literature gap and need for this research immediate before mentioning research objectives 

Methods

Include study setting and clearly mentions about study settings

How researchers accessed data

Inclusion and exclusion criteria of study participants

Researchers mentioned in the abstract that "we analyzed data from 736 pregnant women who had injuries that were an abbreviated injury". However, in the method section, there is not any detail about how the researchers included 736 cases from 141,057 collisions.

Clearly mention the variable in which you choose for this study. The recording procedure of relevant variables. 

Mention who screened the data? 

Collected data

I am not able to understand this section. Clearly mention your screening procedure. Maybe re-write this section for a better understanding of the readers.

I would suggest adding a flow chart diagram to show how many incidents were in this specific time and how you finalize the final sample size. 

Result

What is MIAS in line# 137 and in figure 1. Explain this abbreviation first and then use it. Furthermore, if you are using this abbreviation in figure 1. It is important to write the full name below the figure with the *mark.

Line#154 mention the sample size (n=) with table heading 

Discussion

The discussion section is well prepared and clearly mentioned the strengths and weakness of the study

Conclusions

I would suggest re-writing the conclusions section. Do not repeat your method and some key findings in the conclusions section. In the conclusions section, researchers may give some policy measures on this issue and suggest some recommendations in this section as well.

Author Response

Thank you for the thoughtful and constructive feedback you provided regarding our manuscript. In the following sections, you will find our responses to each of your points and suggestions. We are grateful for the time and energy you expended on our behalf.

Abstract

Neither title nor abstract reporting the site of research. It is important to mention the site in the abstract for a quick understanding of readers. The site can be added in Line#16-17

  Following to your suggestion, we have added the site of research in Abstract as follows (Line 17-18): a retrospective analysis using the National Automotive Sampling System/Crashworthiness Data System was performed in Shiga University of Medical Science”.

 Introduction

It would be great to mention/cite some previous studies that focus on factors influencing pregnant women’s injuries and fetal loss due to motor vehicle collisions. If there is no study on this topic, then clearly mention the literature gap and need for this research immediate before mentioning research objectives

According to your suggestion, following sentences have added in Introduction (Line 60-84): “For patients due to MVCs, there were several factors about the collision characteristics that influenced the mechanisms of injuries and outcomes. Therefore, collision details have to be considered in the study of pregnant woman vehicle passengers who are involved in a collision. Previously, a nationwide hospital-based database was analyzed, and the factors for pregnancy loss or for requiring surgery were determined [11-13]. Additionally, national or regional population-based databases have been used to study pregnant women who were involved in MVCs [3, 14-17]. However, these data-bases lack detailed information on vehicle collisions. Therefore, the NASS/CDS database, which includes information about the crash circumstances and scene, have been used to determine the relationship between the crash severity and pregnant women’s injuries or outcomes. Previously, several studies on pregnant women who were involved in MVCs were performed using this database or similar in-depth investigations [10, 18, 19]. Klinich et al. suggested that a greater crash severity, more severe maternal injuries, and lack of proper seatbelt use were associated with an adverse fetal outcome with in-depth investigations of crashes. However, this study was based on small numbers of collisions involving 57 pregnant occupants, and factors about the principal direction of the force, action of pretensioner system, or rollover were not included in the analysis [18]. Manoogian compared the crash and injury characteristics between pregnant and non-pregnant vehicle occupants and found that the risk of injuries with a AIS score of 2 or more for pregnant occupants was similar to the risk of those for non-pregnant occupants [10]. Collins et al. compared the restraint use rate between pregnant and non-pregnant women who were involved in MVCs, and they concluded that pregnant women wear belts at significantly lower frequencies than non-pregnant women [19]. However, no study has determined the factors that influence more severe injuries for pregnant women and a negative fetal outcome in MVCs where the crash details were included”.

Methods

Include study setting and clearly mentions about study settings

  In accordance with your suggestion, following sentence was added in Material and methods (Line 99-100): “The following analyses were performed in Shiga University of Medical Science”.

How researchers accessed data

In accordance with your suggestion, following sentence was added in Material and methods with additional reference (Line 98-99): “The raw data were downloaded by some of authors (ST and YM) via FDP Site of NASS/CDS in July 2017 [20]”.

Inclusion and exclusion criteria of study participants

Researchers mentioned in the abstract that "we analyzed data from 736 pregnant women who had injuries that were an abbreviated injury". However, in the method section, there is not any detail about how the researchers included 736 cases from 141,057 collisions.

  As the reviewer suggested, the inclusion and exclusion criteria of study participants were obscure. Therefore, we have added the subheading, “Patient selection” and have described the procedure in detail with additional flowchart (Figure 1) as follows (Line 102-112): “A total of 65,390 collisions involving 141,057 persons were registered in the NASS/CDS from 2001 to 2015. Among these collisions, we included cases that involved at least one pregnant woman occupant, subsequently, 1,074 collisions involving 1,088 pregnant women were collected. The 352 pregnant women who had no anatomical injuries indicating a AIS score of 0 were excluded from analyses. Finally, 736 pregnant women having injuries with an AIS score of 1 or more were selected for analyses (Figure 1)”.   

Clearly mention the variable in which you choose for this study. The recording procedure of relevant variables.

  In accordance with your suggestion, we have revised the descriptions in Material and methods as follows (Line 115-120): “The database included the variety of information regarding the involved pregnant women (i.e., age, stature, gestational age, type and severity of injuries) and the collisions (i.e., seating position, direction and velocity of the collision, use or acting of safety systems). From the information, we chose the data considered as influencing on the mechanism of injuries of the pregnant woman. Subsequently, following information was obtained from the database for each victim”.

Mention who screened the data?

  In accordance with your suggestion, following sentence was added in Material and methods (Line 108-109): “This screening procedure was performed by some of authors (MH, SM, ST and YM)”.

Collected data

I am not able to understand this section. Clearly mention your screening procedure. Maybe re-write this section for a better understanding of the readers.

I would suggest adding a flow chart diagram to show how many incidents were in this specific time and how you finalize the final sample size.

  As the reviewer suggested, descriptions were obscure. Therefore, we have divided the subheading for “Patient selection” and “Collected data” and added the flowchart of patient selection (Figure 1).

  We have described the procedure in detail as follows (Line 102-112): “A total of 65,390 collisions involving 141,057 persons were registered in the NASS/CDS from 2001 to 2015. Among these collisions, we included cases that involved at least one pregnant woman occupant, subsequently, 1,074 collisions involving 1,088 pregnant women were collected. The 352 pregnant women who had no anatomical injuries indicating a AIS score of 0 were excluded from analyses. Finally, 736 pregnant women having injuries with an AIS score of 1 or more were selected for analyses (Figure 1)”. 

  Also, following description was added in Collected data (Line 114-120): “The database included the variety of information regarding the involved pregnant women (i.e., age, stature, gestational age, type and severity of injuries) and the collisions (i.e., seating position, direction and velocity of the collision, use or acting of safety systems). From the information, we chose the data considered as influencing on the mechanism of injuries of the pregnant woman. Subsequently, following information was obtained from the database for each victim”.

Result

What is MIAS in line# 137 and in figure 1. Explain this abbreviation first and then use it. Furthermore, if you are using this abbreviation in figure 1. It is important to write the full name below the figure with the *mark.

  In the original version, MAIS was already defined and full name was shown in Material and methods (Line 95-96 in original version). According to your suggestion, full name of MAIS was added in the legend of Figure 2 (Line 185-186).

Line#154 mention the sample size (n=) with table heading

Following to your suggestion, we have added the sample size in the heading of Table 1 as follows (Line 196): Comparison of the background and collision characteristics of pregnant women with a MAIS of 1 (n=568) or 2 or more (n=168).

Discussion

The discussion section is well prepared and clearly mentioned the strengths and weakness of the study

  We appreciate your kind comment.

Conclusions

I would suggest re-writing the conclusions section. Do not repeat your method and some key findings in the conclusions section. In the conclusions section, researchers may give some policy measures on this issue and suggest some recommendations in this section as well.

  Following to your suggestion, we have revised the descriptions of Conclusion as follows (Line 327-332): “To save the fetus, avoiding moderate to severe maternal injuries is a high priority. We health-care professionals should provide proper counselling about correct seatbelt use and safety education to prevent excessive speed for pregnant women. These measures might improve the fetal outcome if pregnant women are involved in MVCs. Subsequently, the measures also promote the social participation of pregnant women with relieving the anxiety of fetal loss”.

Round 2

Reviewer 2 Report

The authors have addressed all the comments and I suggest accepting this manuscript for publication.